# Hydrotreated Vegetable Oil as a Fuel from Waste Materials

**Petr Zeman [1], Vladimír Hönig [1,\*], Martin Kotek [2]** **, Jan Táborský [1], Michal Obergruber [1], Jakub Mařík [2], Veronika Hartová [2] and Martin Pechout [2]**

[1]    Faculty of Agrobiology, Food and Natural Resources, Department of Chemistry, Czech University of Life Sciences Prague, Kamýcká 129, Prague 6, 169 21, Czech Republic; pzeman@af.czu.cz (P.Z.); taborsky@af.czu.cz (J.T.); obergruber@af.czu.cz (M.O.)

[2]    Faculty of Engineering, Department of Vehicles and Ground Transport, Czech University of Life Sciences Prague, Kamýcká 129, Prague 6, 169 21, Czech Republic; kotekm@oikt.czu.cz (M.K.); marikj@tf.czu.cz (J.M.); nidlova@tf.czu.cz (V.H.); pechout@tf.czu.cz (M.P.)

\*    Correspondence: honig@af.czu.cz; Tel.: +420-22438-2722

**Abstract:** Biofuels have become an integral part of everyday life in modern society. Bioethanol and fatty acid methyl esters are a common part of both the production of gasoline and diesel fuels. Also, pressure on replacing fossil fuels with bio-components is constantly growing. Waste vegetable fats can replace biodiesel. Hydrotreated vegetable oil (HVO) seems to be a better alternative. This fuel has a higher oxidation stability for storage purposes, a lower temperature of loss of filterability for the winter time, a lower boiling point for cold starts, and more. Viscosity, density, cold filter plugging point of fuel blend, and flash point have been measured to confirm that a fuel from HVO is so close to a fuel standard that it is possible to use it in engines without modification. The objective of this article is to show the properties of different fuels with and without HVO admixtures and to prove the suitability of using HVO compared to FAME. HVO can also be prepared from waste materials, and no major modifications of existing refinery facilities are required. No technology in either investment or engine adaptation of fuel oils is needed in fuel processing.

**Keywords:** biofuel; biodiesel; hydrotreating; hydrocarbon; waste

## 1. Introduction

A long-term European strategy is an effort for a so-called "recycling society". With the growing volume of waste, expanding industries are dealing with waste management and recycling. Despite noticeable progress, there is still great potential in previously underutilized sources of waste. The main obstacles are the legislative problems and the low application of approved rules, the differentiation of regulations in different countries, and generally the low awareness of the professional and lay public about new possibilities and prospects. The current EU waste policy is based on the concept of the so-called waste hierarchy, which states that it is primarily necessary to prevent the generation of waste itself, and if this is not possible, it must be recycled or otherwise exploited, under condition of minimal dump disposal. Anything which can be reused in some way may be considered waste, even materials like grey water, wood chips, old clothing, kitchen scraps or diseased fruit and vegetables [1–7].

Legislative requirements are higher for double-counting materials, in order to meet the 10% share of biofuels. This double counting applies to biofuels made from waste and residues, as well as to biofuels made from raw materials that have been grown on so-called degraded areas, and it is thus another supportive step towards meeting the sustainability criteria [8–10].

Sustainability criteria in the EU are determined by Directive 2009/28/ES. Among these, we include reducing GHG (greenhouse gas) emissions, optimizing land use and carbon stocks, biodiversity, and

environmental requirements for crop production. To achieve 10% $CO_2$ savings between 2010 and 2020, a minimum biofuel share of 15% must be reached.

Hydrotreated vegetable oils are one of the possible ways of using the increased biofuel content in diesel fuel. This would provide another option for meeting the $CO_2$ reduction target for the year 2020. At the same time, it will be necessary to include these advanced biofuels in legislation, and to establish clear rules for their use. In addition to these changes, changes in the composition of diesel fuel can also be expected in the future. This mainly concerns requirements for increasing the cetane number and cetane index, adjusting the course of the distillation curve (reduction of temperature by 95% of the pre-distilled volume), further reducing the content of polycyclic aromatic hydrocarbon, introducing a limit for aromatics similar to automotive gasoline and tightening requirements for lubrication and mechanical impurities for fuels for diesel engines. Introducing changes that have a positive effect on reducing harmful emissions and particulate matter pollution will entail an increase in production costs and therefore the speed of their deployment will depend on the economic situation and legislative changes adopted within the EU [11–13]. Compared to fossil fuels, biofuels are renewable. As far as their technological development is concerned, the issue of biofuels is only at an early stage. The most commonly declared "first generation" of biofuels is bioethanol produced from starch and sugar, biodiesel produced from vegetable oils (rape, soy, etc.) and animal fats without chemical treatment, or produced due to the transesterification process to fatty acid methyl esters (FAME—rapeseed oil) or similar non-edible feedstocks like soursop seed oil [14,15]. These are sophisticated technologies, and above all, commercially available [16–18].

Figure 1 shows a simplified difference between the production of hydrotreated oils and fatty acid methyl esters (FAME).

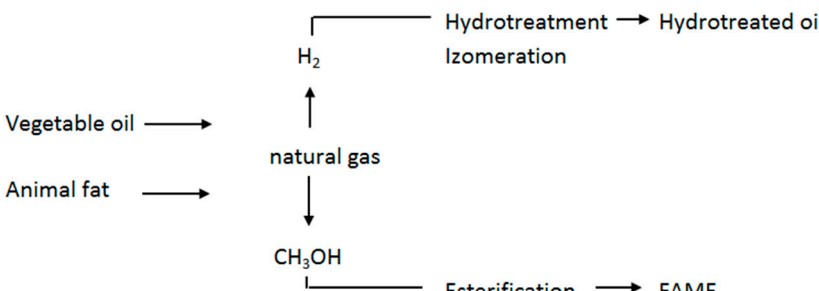

**Figure 1.** Simplified diagram for the production of hydrotreated oils and fatty acid methyl esters. FAME: fatty acid methyl esters.

The production of hydrotreated vegetable oils is based on introducing hydrogen molecules into the raw fat or oil molecule. This process is associated with the reduction of the carbon compound. When hydrogen is used to react with triglycerides, different types of reactions can occur, and different resultant products are combined.

The original oil obtained by hydrotreatment achieves higher oxidation stability, which is desirable for frying oils. Partial fat stiffening is used for raw margarine production. For fuel purposes, such a final product is not suitable. In hydrotreated fuels, therefore, partial hydrotreatment is mostly omitted and overall hydrotreatment continues, often with free fatty acids (Figure 2).

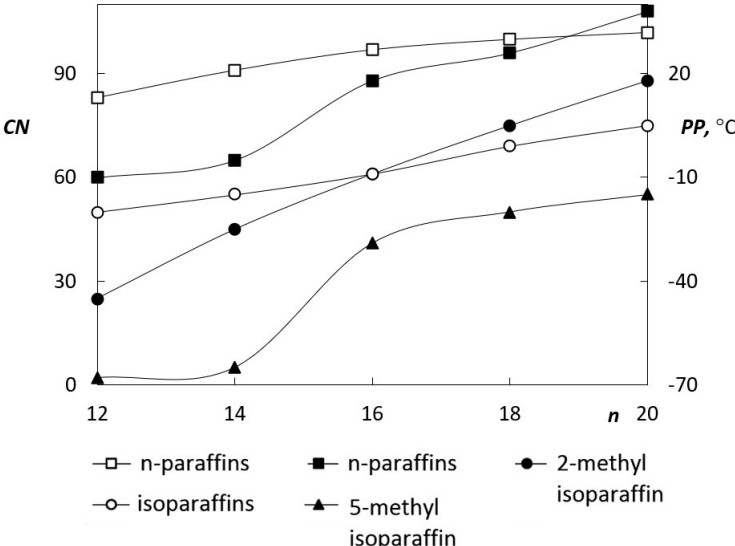

**Figure 2.** Hydrotreatment of fatty acids.

Another method of converting triglycerides by hydrogen is the cleavage of the ester to hydrocarbon and glycerol-derived propane and free fatty acids. These fatty acids (n is the number of carbons) are either:

1.  reduced to hydrocarbons (n) and water by hydro deoxygenation,
2.  subject to decarboxylation, i.e. carbon dioxide $CO_2$ is cleaved to give n-1 hydrocarbons,
3.  or decarboxylation is carried out by removing the carbon monoxide (CO) and water molecule to produce an n-1 hydrocarbon.

For the reaction of hydrogen with vegetable oils and vacuum gas oil, the same catalysts and the same types of reactors and equipment were used as for the oil processing [19,20]. In recent decades, efforts have been made to find the best catalysts, to optimize hydrogen reaction processes and to find suitable sources of vegetable oil or fats. A lack of resource availability and high hydrogen consumption are increasing production costs, but these shortcomings are being gradually managed commercially [21].

A key process for obtaining hydrocarbons is hydroisomerization. This is a radical reaction where branching of hydrocarbon molecules is achieved by the use of form-selective catalysts, such as zeolites or other acid catalysts. N-paraffins having a boiling point corresponding to diesel fuel generally have a higher cetane number than their branched isomers. In contrast, isoparaffins have lower solidification points than n-paraffins. Therefore, there is a compromise in the quality of the paraffin-rich fuel; the fuel has either good combustion properties, or good low-temperature properties. The result of hydroisomerization is therefore a fuel with a lower solidification point and a lower cetane number. The relationship of these two properties is illustrated in Figure 3 [22].

**Figure 3.** Cetane number and paraffin solidification points depending on carbon number.

Hydrotreated oils are characterized by very good low temperature properties. The cloud point also occurs below −40 °C. Therefore, these fuels are suitable for the preparation of premium fuel with a high cetane number and excellent low temperature properties. The cold filter plugging point (CFPP) virtually corresponds to the cloud point value, which is why the value of the cloud point is significant in the case of hydrotreated oils.

The production process of hydrotreated oils can also produce fuel with appropriate low temperature properties from palm oils and other oils, including animal fats, whose methyl esters have a very poor applicability at lower temperatures [23]. This results in the use of hydrotreated oils throughout the year, without risking loss of serviceability or fuel logistics problems. Hydrotreated vegetable oils thus find their potential usability as aviation fuels.

Hydrotreated vegetable oils meet EN 15940:2014 for paraffinic diesel fuels from synthesis or hydrotreatment, formerly TS 15940:2012 for paraffinic diesel fuels [24]. This specification also applies to Fischer-Tropsch synthesis products: GTL, BTL, and CTL. Specification TS 15940:2012 was preceded by CWA 15940:2009 CEN Workshop Agreement, which was created in cooperation between car and fuel manufacturers. HVO is usually supplied without FAME, however, it is allowed to add up to 7% vol. under specification EN 15940, which the earlier CWA 15940 did not allow. EN 14214 for FAME for HVO does not apply, as HVO is composed only of hydrocarbons. However, HVO meets the requirements of EN 590 for diesel fuel, except for the density below the lower limit of this standard [25]. This also applies to the US ASTM D975 [26]. Table 1 shows the differences between these standards.

**Table 1.** Requirements of EN 15940, EN 590 and ASTM D975 [27].

| Parameter | EN 15940 | EN 590:2013 | ASTM D975 |
|---|---|---|---|
| Cetane number | ≥70.0 | ≥51.0 | ≥40 |
| Density at 15 °C (kg·m$^{-3}$) | 765–800 | 820–845 | |
| Viscosity at 40 °C (mm$^2$·s$^{-1}$) | 2.00–4.50 | 2.00–4.50 | 1.9–4.1 |
| Hydrocarbons (% m/m) | | | ≤35 |
| Polyaromatic | - | ≤8 | - |
| Aromatic | ≤1.0 | - | - |
| Olefin | ≤0.1 | - | - |
| Sulfur content (mg·kg$^{-1}$) | ≤5.0 | ≤10.0 | ≤15 |
| Flash point (°C) | >55 | >55 | >52 |
| Lubricity HFRR at 60 °C (μm) | ≤460 * | ≤460 | ≤520 |
| 95% by volume distils at (°C) | <360 | <360 | 282–338 |
| CFPP (°C) | ≥−34 | ≥−34 | - |
| Ash content (% m/m) | ≤0.01 | ≤0.01 | ≤0.01 |
| Total impurity content (mg·kg$^{-1}$) | ≤24 | ≤24 | - |

* Including lubricating additives for use in vehicles approved for driving on the fuel according to the standard.
CFPP: cold filter plugging point; HFRR: high frequency reciprocating rig.

## 2. Results

The density of the hydrotreated vegetable oil (about 780 kg·m$^{-3}$) is lower than the density of fossil diesel (800 to 845 kg·m$^{-3}$) because of its paraffinic character, and also a lower temperature distillation end. The density of the fuel has traditionally been an important factor in terms of volume of fuel consumption and maximum performance, and we can say that the reduction of volume calorific value is only a function of density. At a lower calorific value, the engine generates less energy and needs more fuel to ensure the same power output at a partial load.

In the case of hydrotreated oils, the effect is different, as the calorific value compensates for the lower density effect (Figures 4 and 5). The higher calorific value of the hydrotreated oils is due to the fact that the amount of hydrogen in the hydrotreated oils is about 15.2% (m/m), as opposed to 13.5% (m/m) in standard diesel.

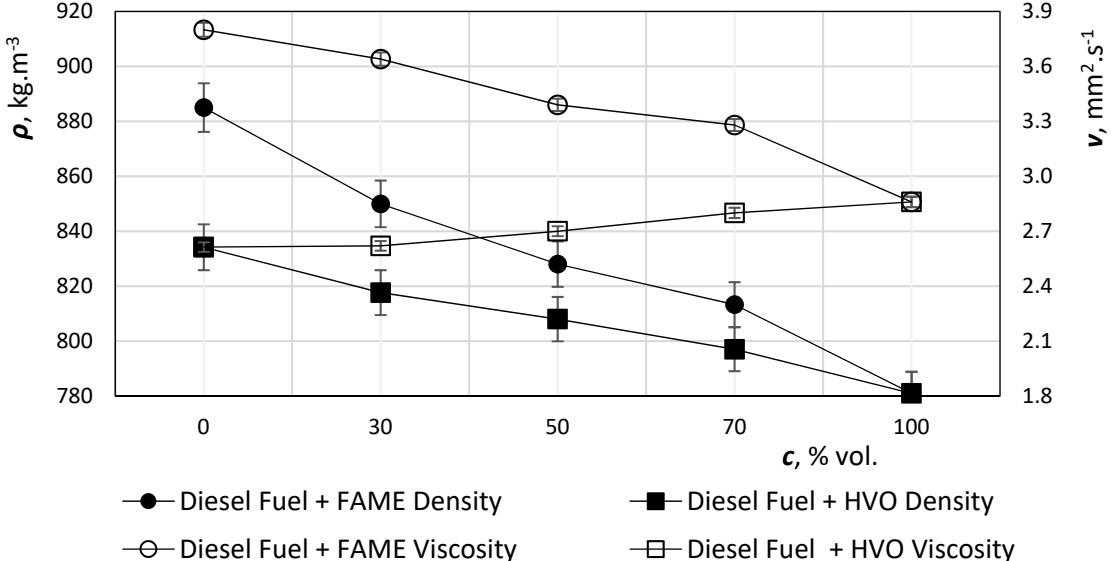

**Figure 4.** Density and kinematic viscosity of diesel fuel with FAME and hydrotreated vegetable oil (HVO).

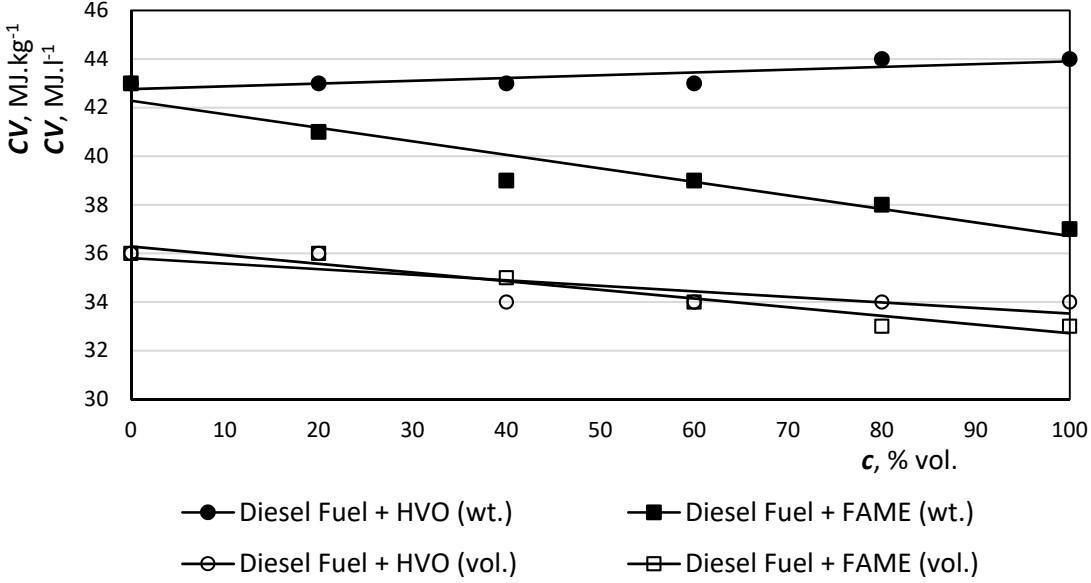

**Figure 5.** Calorific value of mixed fuels.

Figure 4 shows that the density with increasing HVO concentration in the mixture is expected to decrease. This is due to the lower permissible water content and paraffinic character with a higher hydrogen content than diesel, which results in a higher energy content per kg. Mixture of diesel fuel with HVO > 30% vol. exceeds the EN 590 (Table 1) limit for diesel ($820$–$845$ kg·m$^{-3}$). High proportion mixtures did not meet the standard limits. On the contrary, low density offers the possibility of blending HVO into diesel fuels with higher contents of heavier fractions, or their incorporation into less profitable products, such as fuel oil. Conversely, all HVO mixtures meet standardized kinematic viscosity limits.

On the *y*-axis, lubricity (WSD—wear scar diameter) is in μm. HVO has very low lubricity, therefore up to 80% of the volume may be added as maximum. Once the concentration reaches 80% or more, the mixture of the fuel does not match the EN standard—see Figure 6. It could lead to seizure of the fuel system of the machine.

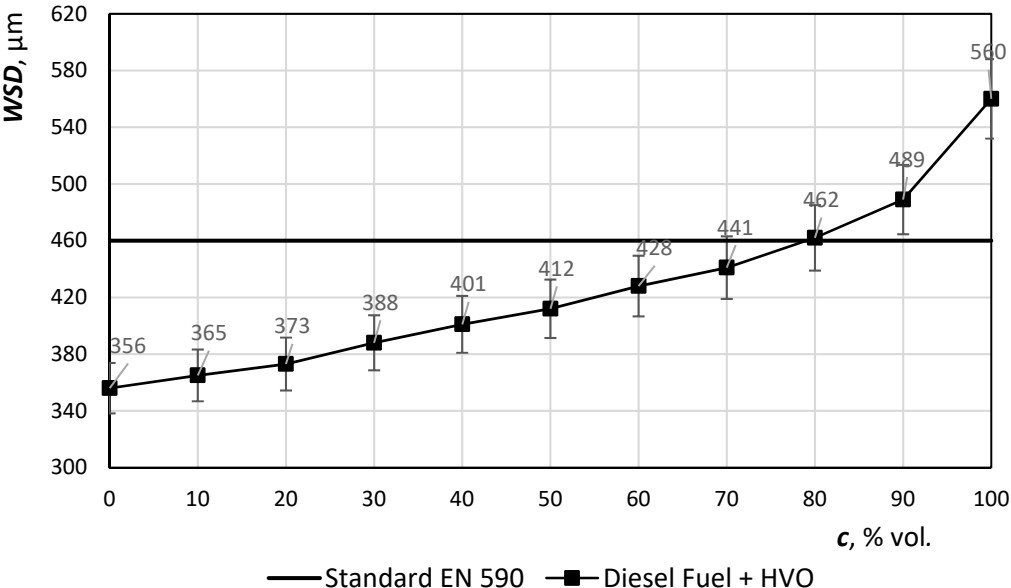

**Figure 6.** Lubricity of diesel fuel with addition of HVO.

An aromatic-free hydrocarbon composition results in a lower lubricity of the fuel. Lubrication of hydrotreated oils corresponds to sulfur-free winter grade diesel or GTL. It is essential that lubricant additives are added to these fuels to meet the requirements of EN 12156-1 (HFRR high frequency reciprocating rig, corrected abrasion area diameter at 60 °C < 460 mm). It is possible to apply commonly used lubricants for diesel with a similar dosage. When using this fuel at higher concentrations, it is assumed that a further test for lubricity verification will be added.

Hydrotreated oil can also be supplied with lubricating additives for use in pure form, or as an additive. It is common, however, to supply, for example, HVO without these additives if the fuel is designated as a component of the mixture. The lubricity of the resulting fuel must then be controlled and must be increased, to cover the HVO in the mixture. From the point of view of lubricity, this is the only parameter where FAME is better and in itself can replace the additive in all hydrocarbon fuels. The lubricating ability can be improved by adding the appropriate additive or a small amount of FAME content. The expanded uncertainty of the result is ± 5%.

Also, the distillation curve is different from diesel fuel and FAME. Distillation properties show how the fuel evaporates when injected into the combustion chamber of the diesel engine. Low-boiling fractions are important for the engine's start-up, and the heavier fractions with higher boiling points can cause problems with fuel being incompletely burned and increasing the proportion of harmful emissions in the engine exhaust gases. Standard diesel fuel has a boiling range approximately from 180 °C to 360 °C.

A distillation test to determine the distillation curve is a test that must always be carried out when assessing the quality of fuel. The distillation curve expresses the volume percentage of the fuel that is distilled to a certain distillation temperature. T50 is the temperature at which 50% of the fuel is distilled. At this point the amount of air is bound for perfect combustion. For HVO and diesel, there is no need to worry about combustion air as the temperature in T50 has not increased significantly; the value is shown in Figure 7.

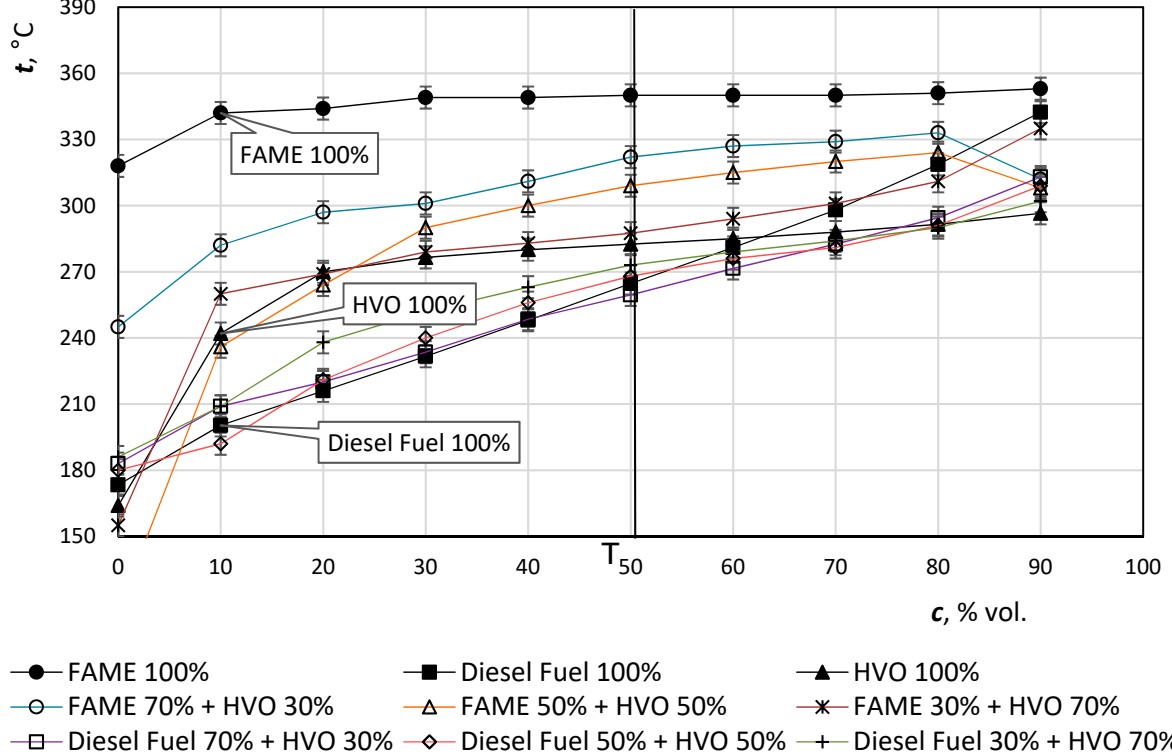

**Figure 7.** Distillation curves of FAME, diesel fuel, HVO and their mixtures.

It is clear from Figure 7 that HVO does not affect the beginning of fuel distillation. The presence of "light components" is not compromised, so the moving parts of the fuel system cannot be damaged. The addition of HVO results in a flattening of the distillation curve. Distillation indicates a lower proportion of high boiling heavy fractions, thereby reducing carbonization shares and reducing exhaust emissions. Higher concentrations of HVO can be expected to affect engine performance.

Figure 8 shows the difference of values due to the HVO admixtures. The addition of HVO to diesel fuel positively affects the loss of filterability (CFPP—cold filter plugging point). Standard EN 590 sets the temperature −20 °C as the maximum value for F-class winter diesel, 14 is marked with a thick horizontal line.

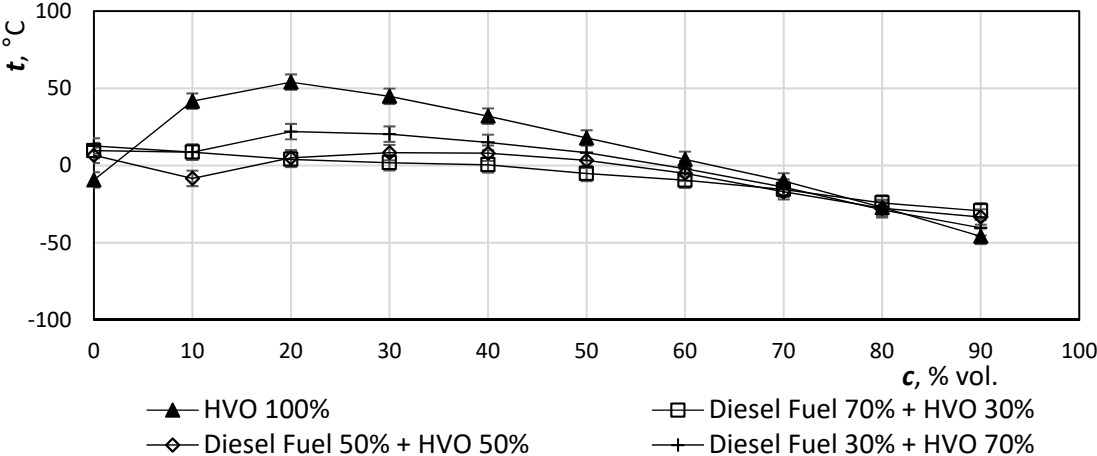

**Figure 8.** Changes in the mixed fuel distillation curve as compared to pure mineral diesel on the *x*-axis.

The cloud point (CP) or wax appearance temperature (WAT) is the temperature at which n-paraffins begin to precipitate in fuel, but generally, it is not a mandatory indicator.

Figure 9 illustrates the decrease of temperature with increasing addition of HVO. In the case of 100% HVO, the CP value is practically the same as the CFPP value. HVO addition to diesel fuel favors low temperature properties, which, even in the case of an HVO 30 mixture, are 6 degrees below the winter diesel fuel EN Class F. EN 590 also sets the CFPP for an arctic climate (for 1st class it is −26 °C, while for 2nd class it is −32 °C), both of which exceed FAME + HVO 70−100.

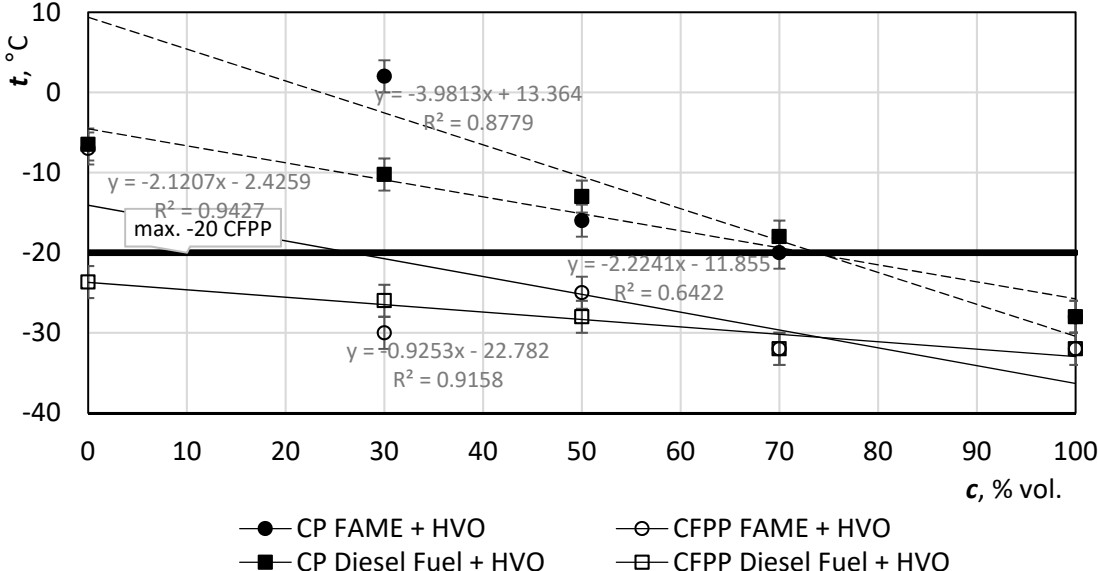

**Figure 9.** CFPP and cloud point (CP) of mineral diesel fuel with FAME and HVO.

At temperatures above the cloud point, the hydrotreated oil is colorless, clear as water. It has no characteristic aroma typical for other fuels. It does not contain any visible dirt at temperatures above the cloud point. The cloud point causes the creation of scum characteristic for diesel.

The flash point is the lowest temperature at which the flammable substance produces so many flammable vapors at atmospheric pressure that they will briefly ignite being in contact with open flame but they do not continue to burn.

Figure 10 shows the temperature increase of the flash point with increasing HVO content. Temperature range III of the hazard class, in which the measured temperatures of all HVO mixtures are located, is defined here. However, in the case of a flash point value, these values do not affect the work of the diesel engine.

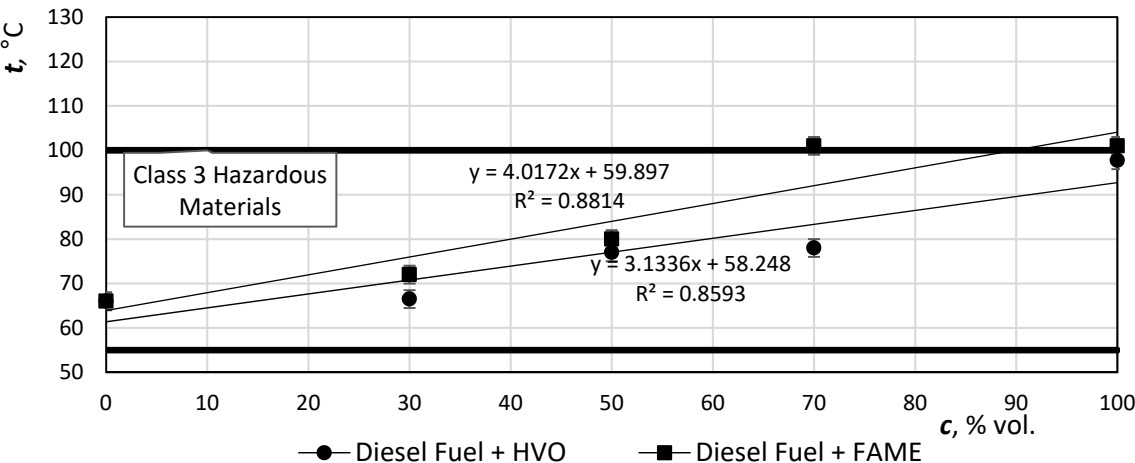

**Figure 10.** Flash points of diesel fuel with HVO and FAME.

The cetane number indicates the reactivity of diesel fuel in terms of its diesel characteristics. The higher the fuel gets, the better it is, the higher the cetin number is, the more regular and better is its combustion, as well as the engine running and noise. Because of the relative difficulty of the cetane number test, the cetane index, which can be determined based on a calculation from the results of the laboratory density and distillation tests, has been introduced as a characteristic of ignition capability. According to EN 590, the cetane number is at least 51 units, the cetane index is 46 units.

Figure 11 shows the increasing trend lines of the cetane number and cetane index with increasing HVO content. The cetane number of hydrotreated vegetable oils ranges from 75 to 95 units due to the composition (n-paraffins and isoparaffins). In mixed fuels, there is a linear increase in the cetane number, corresponding to the proportion of components. Hydrotreated oil is a suitable additive for increasing the cetane number due to the nature of the fuel, where its effect is greater than the use of conventional additives. For measuring the cetane number on the test engine, the hydrotreated oil must be mixed with a fuel with a known and low cetane number, such that the cetane number of the resulting mixture is below 70 units within the measuring range. Then, the cetane number of the hydrotreated vegetable oil is determined by linear extrapolation. The calculation of the cetane index is suitable for standard diesel fuels (with FAME) and its use for hydrotreated oils is not appropriate.

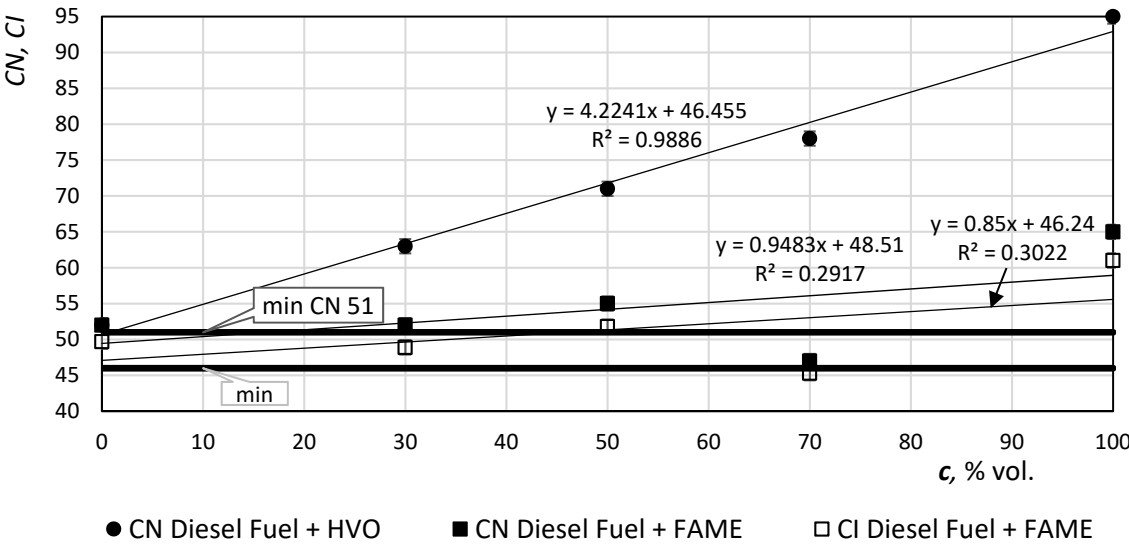

**Figure 11.** Cetane numbers of diesel fuel with HVO and FAME, cetane index of diesel fuel with FAME.

The limit value for the cetane number according to EN 590 is highlighted in the graph with a horizontal line; for the cetane index, 46 is also highlighted by the horizontal line. The extrapolated cetane number is high because of the very high content of n- and isoparaffins in HVO; the value of the HVO 30 mixture already means a much higher cetane number than the minimum value according to EN 590.

The measurement in this work compared mineral diesel fuel without bio-components, 100% pure HVO and their mixtures. Prepared ratios were based on the possibility of comparison of the results of our own measurements and results of measurements already published in the literature. The individual measurement procedures were performed according to applicable valid standards and were repeated three times to avoid any measurement error. The density measurement results of mixtures with increasing HVO concentration had an expected decreasing trend, which should have a negative effect on the calorific value (energy content per kg). According to the HVO manual, this drop is due to the lower permitted water content and the pure paraffinic character of HVO fuel, which mineral diesel does not have, but due to its higher hydrogen content, HVO has a higher calorific value at lower density.

Since hydrotreated oil consists only of hydrocarbons, traditional methods for fossil diesel, but not FAME, are also suitable for determining fuel stability. This especially applies to "Rancimat" methods according to EN 15751, which is intended for pure diesel fuel and FAME containing 2–7% vol. of FAME. This method is not suitable for pure hydrotreated oil, even as an additive in diesel fuel. The stability of hydrotreated oils is at the standard level of diesel fuel and there should be no risks except for long-term vehicle shutdown or storage.

The sulfur content of hydrotreated oil is based on the production process and is <1 mg·kg$^{-1}$. As the standard oil logistics system is used for hydrotreated oil, the sulfur content due to contamination may be higher, and then the normalized value is ≤5.0 mg·kg$^{-1}$. Addition of hydrotreated oils can also positively reduce the sulfur content, for example in diesel, where the value exceeds the relevant standard EN 590.

The ash content in hydrotreated oils is very low (<0.001%). Also, the content of P, Ca and Mg is well below the detection limits of analytical methods (<1 mg·kg$^{-1}$).

Hydrotreated oil is, like fossil hydrocarbons, nonpolar, while water is polar. Water solubility is thus similar to traditional diesel fuel, or even lower. Therefore, the issue of water requires no further action in the field of logistics (as well as for diesel we use).

The issue of microbial growth is primarily about FAME, which also promotes microbial growth in diesel fuel blends. FAME is biodegradable and tends to increase the water content of diesel fuel. Unlike FAME, the presence of hydrotreated oil mixed with diesel fuel does not require any further action. However, monitoring the quality indicator is useful because microbes can proliferate even in pure fossil fuel during long term storage in the presence of free water. Higher temperatures, especially in the summer, can increase microbial growth, mainly if mineral salts are present in the water phase. At lower temperatures, growth of microorganisms slows down.

The hydrocarbon composition of HVO corresponds approximately to the hydrocarbons of which diesel fuel is composed. The composition of HVO is composition closer to diesel oil than to a FAME mixture, which is an advantage for the use of HVO as a substitute for FAME.

R is detector response in Volts; $t_R$ is retention time in minutes. Figures 12–14 are chromatograms of fuel samples (diesel fuel, HVO, and FAME). In Figure 12, we can see the composition of analyzed diesel fuel with labeled n-alkanes, which represent major constituents of the sample. Similarly, Figure 13 shows the composition of hydrocarbon compounds in HVO. The chromatogram in Figure 14 allowed us to specify the presence of major components (methyl esters) in FAME; the peaks are well separated. All components labeled in the chromatograms were identified according to retention times of analytical standards (mixture of n-alkanes), which are specified in the section Materials and Methods.

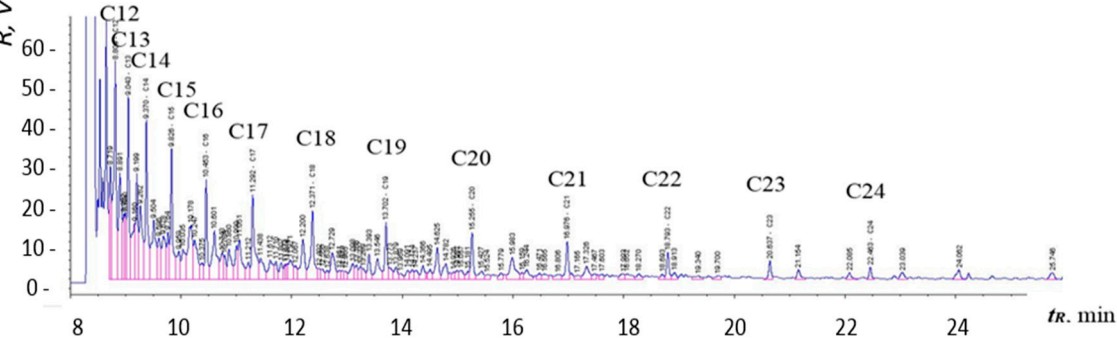

**Figure 12.** Chromatogram of 100% diesel fuel with identified (labeled) n-alkanes.

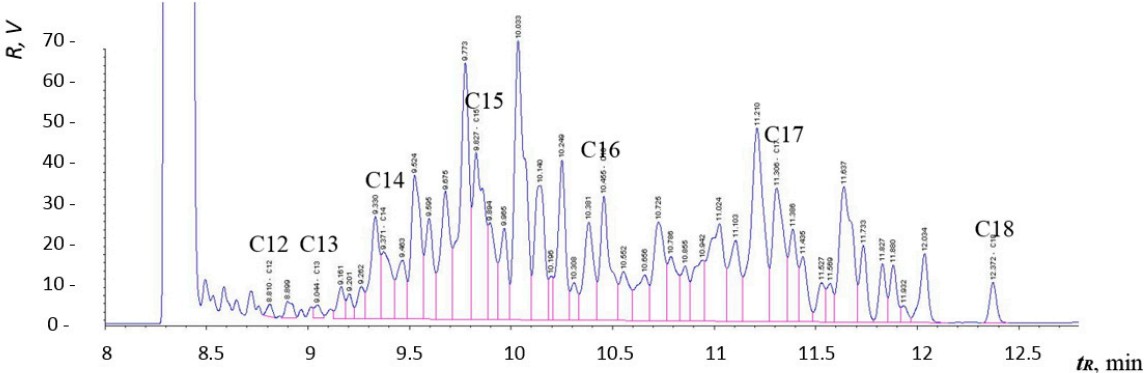

**Figure 13.** Chromatogram of 100% HVO with identified (labeled) n-alkanes.

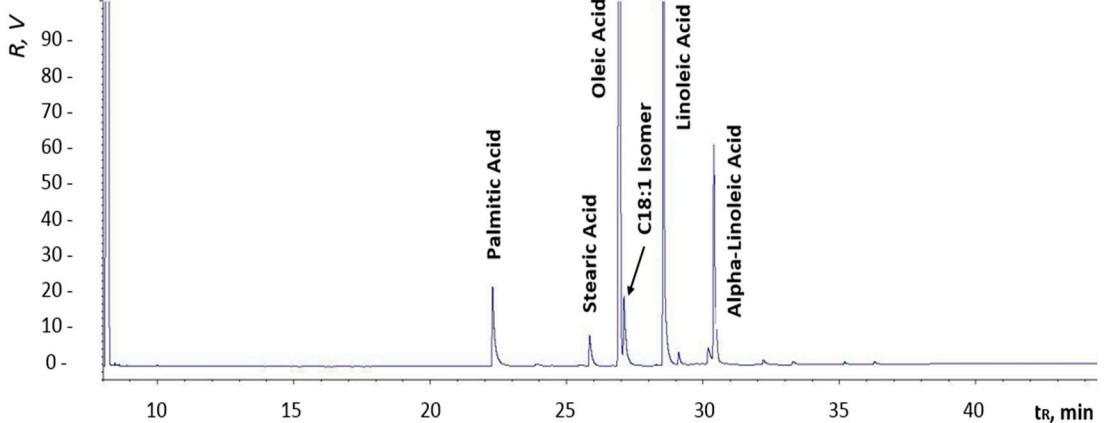

**Figure 14.** Chromatogram of 100% FAME with identified main components (methyl esters).

Chromatographic results amongst individual samples may vary, depending on the refinery they come from. Generally, the final composition of the fuel depends on the season, the country of origin, the class of fuel, and more.

A statistical analysis was then conducted to obtain a general equation of density and viscosity for independent variable concentration. An analysis was done for a mixture of diesel and HVO, and a mixture of FAME and HVO. For the needs of the article, statistical tool R, and the built-in library lm() were used.

As a first analysis, a linear regression analysis was calculated to predict the density of a diesel and HVO mixture based on concentration. A significant regression equation was found ($F(1, 3) = 1411$, $p < 0.01$, $N = 20$), with $R^2$ of 0.9972. The model predicted density in the form of equation (1). Assumptions of linear regression were verified by gvlma library. All assumptions were accepted: Global Stats ($p = 0.5704$), Heteroscedasticity ($p = 0.9905$), Skewness ($p = 0.2229$), Kurtosis ($p = 0.9911$), and Link Function ($p = 0.2302$).

$$\rho = 0.52147c + 780.97672 \tag{1}$$

Complementary linear regression was calculated to predict the viscosity of a diesel and HVO mixture based on concentration. A significant regression equation was found ($F(1, 3) = 49.97$, $p < 0.01$, $N = 20$), with $R^2$ of 0.9245. The model predicted viscosity in the form of Equation (2). All assumptions were accepted: Global Stats ($p = 0.5884$), Heteroscedasticity ($p = 0.8937$), Skewness ($p = 0.7208$), Kurtosis ($p = 0.4098$), and Link Function ($p = 0.1579$).

$$v = -0.003875c + 2.89085 \tag{2}$$

As a second analysis, a linear regression analysis was calculated to predict the density of a FAME and HVO mixture based on concentration. A significant regression equation was found ($F(1, 3) = 94.09$,

$p < 0.01$, N = 20), with $R^2$ of 0.9588. The model predicted density in the form of equation (3). Assumptions of linear regression were verified by gvlma library. All assumptions were accepted: Global Stats ($p = 0.5035$), Heteroscedasticity ($p = 0.8260$), Skewness ($p = 0.4299$), Kurtosis ($p = 0.1330$), and Link Function ($p = 0.5241$).

$$\rho = 0.7478c + 785.6795 \tag{3}$$

Complementary linear regression was calculated to predict the viscosity of a FAME and HVO mixture based on concentration. A significant regression equation was found (F(1, 3) = 88.94, $p < 0.01$, N = 20), with $R^2$ of 0.9674. The model predicted viscosity in the form of equation (4). All assumptions were accepted: Global Stats ($p = 0.8565$), Heteroscedasticity ($p = 0.6979$), Skewness ($p = 0.5792$), Kurtosis ($p = 0.4987$) and Link Function ($p = 0.5205$).

$$v = 0.011987c + 2.887347 \tag{4}$$

## 3. Discussion

The experiment shows that the hydrotreatment process is an alternative to the production of biofuels for the esterification process to eliminate the undesirable effects, as described in [16] and [28]. These include, in particular, increased $NO_x$ content, emissions, fuel storage problems, engine oil wear, and so on. HVO are also characterized by high cetane numbers, as confirmed by the measured results [16]. The characteristics for diesel compared to HVO were practically the same, as illustrated by Sugiyama [17]. Experiments and proven measurements show that HVO impurities have a positive effect on the characteristics of diesel engines.

Characteristics to be monitored include, above all, the lubricity to provide the lubricating ability of the moving parts of the fuel system and the cetane number. The graphs show, in accordance with [28], that the recommended HVO ratio (addition) should be about 50% in order to be consistent with the diesel fuel characteristics.

According to Šimáček et al. [29], low density and low sulfur content have an effect on lower lubricity, which can be improved by the application of conventional lubricating additives as is the case with today's low-grade mineral diesel fuel. The kinematic viscosity of all HVO mixtures meets the standard parameters. The distillation curve determines that, by addition of HVO to mineral diesel fuel, its process is flattened. According to Hönig et al. [23], this has a positive effect on the reduction of carbon deposits and exhaust emissions. The HVO Manual [27] indicates CFPP up to −40 °C. This value has not been confirmed by its own measurement. The lowest measured CFPP was −36 °C in 100% HVO. This is even 11 °C less than that found in Aatola et al. [16]. Even with this mismatch, all blends have a positive effect on the CFPP drop and are well below the F-class for diesel fuels, the CFPP reported by EN 590 max −20 °C. Results of the flash point measurement have an increasing tendency, which corresponds to all other articles, the measured values were compared with. This has a positive effect on reducing the risk of a fuel explosion during handling and storage under standard conditions. The measured high values of the cetane number and the calculated cetane index value increase with the HVO content in the mineral oil mixture. The values of the cetane number and cetane index set out in this work correspond to the already high values of these figures in Aatola et al. [16], Šimáček et al. [29], Hönig et al. [30], Váchová and Vozka [31], and the HVO guidelines [27].

The properties of hydrotreated oils are much more similar to high quality sulfur-free diesel or synthetic GTL diesel fuel than to FAME.

In the production of fuels, components with n-hydrocarbons and branched hydrocarbons are suitably combined to achieve suitable fuel properties (cetane number, pour point). Biodiesel produced by the hydrotreating of vegetable oil consists mainly of $C_{17}$ and $C_{18}$ n-hydrocarbons with a high cetane number but with poor low temperature properties due to the melting point between 20 and 28 °C. Improvement of these parameters can be achieved by adding a second proportion of highly isomeric hydrocarbons into the fuel blend.

During long-term storage, pure hydrotreated oils as well as mixtures containing them, behave like traditional diesel fuels. Hydrotreated oils do not contain any hazardous impurities, such as saturated monoglycerides present in FAME. There is therefore no risk of clotting above the cloud point. However, as with standard diesel, this phenomenon may occur due to the presence of paraffins in the fossil fuel or hydrotreated oil during long-term storage at temperatures below the cloud point.

## 4. Materials and Methods

The sample of tested hydrotreated vegetable oil was received from the Neste Oil Company (Espoo, Finland). Simultaneously, diesel fuel free of fatty acid methyl esters compliant with EN 590 and a FAME mixture compliant with EN 14 214 were used for laboratory tests.

The following tests of blends were carried out:

1. Density at 15 °C according to EN ISO 3675
2. Kinematic viscosity at 40 °C according to EN ISO 3104
3. Cold filter plugging point (CFPP) according to EN 116
4. Flash point according to EN 2719
5. Oxidation stability of vegetable oil according to EN 15 751
6. Cetane number according to EN ISO 5165
7. Lubricity according to EN 12156-1 (HVO lubrication 460–650 μm)
8. Calorific value according to ISO 1928 on IKA C200 Calorimeter
9. Gas Chromatography—Flame Ionization Detector GC-FID

The following samples and their mixtures were analyzed:

1. 100% FAME
2. 100% HVO (from Neste Oil Company)
3. 100% Diesel Fuel (without any FAME)

For the GC measurements, the samples were diluted 1/50 (20 μL sample + 980 μL hexane). Analytical Standards:

1. Mixed standard: n-alkanes C10 to C30 in hexane
2. Mixed standard: Supelco 37 Components FAME Mix

For sample analysis, an Agilent Technologies 7890A gas chromatograph (Santa Clara, CA, USA) equipped with an autosampler, a fused-silica capillary column SPB-2560 and a flame ionization detector (FID) was used. The basic instrument parameters and GC analysis conditions are shown in Table 2.

**Table 2.** Device parameters and GC analysis conditions.

| Device | Type and Settings |
| --- | --- |
| Gas Chromatograph | Agilent Technologies 7890A |
| Autosampler | G4513A (16 positions) with a syringe Agilent Gold Standard 10 μL |
| Analytical column | SP-2560, 100 m × 0.25 mm i.d., film thickness 0.2 μm |
| Temperature program | 140 °C (5 min), increase 4 °C·min$^{-1}$, 245 °C (20 min) → 51.25 min |
| Carrier gas | Helium 5.6, const. inlet pressure 50 psi (flow rate 1.58 mL·min$^{-1}$ at 140 °C) |
| Injection chamber | Temperature 280 °C, injection volume 1 μL, split ratio 1:100 |
| Detector | Temperature 280 °C, gas flow: hydrogen (6.0) 30 mL·min$^{-1}$, air (5.0) 400 mL·min$^{-1}$, makeup = nitrogen (6.0) 25 mL·min$^{-1}$ |
| Data collection software | Agilent ChemStation (Revision B.04.02 SP1) |

Lubrication was measured on a PCS instrument: HFRR (high frequency reciprocating rig). The PCS instrument uses an electromagnetically vibrating moving body with low amplitude, while simultaneously compressing it against a solid body. The instrument measures the frictional forces

between the bodies and the electrical contact resistance between them. Settings of the instrument are in Table 3.

**Table 3.** Technical parameters of HFRR.

| Parameter | Value |
|---|---|
| Frequency | 10–200 Hz |
| Shift | 20 μm–2 mm |
| Load | 0.1–1 kg with supplied weights |
| Maximum fractional force | by amplitude, max 10 N |
| Temperature | From room temperature to 150 °C |
| Standard upper test body | Ball ∅ 6 mm |
| Standard lower test body | Disk ∅ 10 mm and thickness 3 mm |
| Power supply | 100–230 V |
| Heating | Two heating cartridges 24 kW, 15 kW |

## 5. Conclusions

Due to the pressure of the European Union to reduce the total amount of greenhouse gases released into the atmosphere, there is also a need to reduce greenhouse gas emissions in transport. One way is to increase the share of biofuels in mineral diesel over 7%.

Biodiesel from FAME is not very suitable as a higher percentage mixture because of the low oxidation stability, higher temperature of cold filter plugging point (CFPP), carbonization tendency and possible microbial contamination in the presence of water. A much better biofuel is HVO, whose hydrocarbon character can be compared to high quality mineral diesel with a very high cetane number and a very low temperature of cold filter plugging point (CFPP). As confirmed by the actual measurement, HVO does not have the above-mentioned drawbacks as fatty acid methyl esters do. HVO can be mixed into mineral diesel fuel without limitation. Its presence in mineral oil blends improves engine performance and reduces fuel consumption, exhaust emissions, and cold filter plugging point (CFPP), so it can also be used in aviation turbine engines. The properties of hydrotreated oils are much more similar to high quality sulfur-free diesel or synthetic GTL diesel fuel than to FAME.

Adding HVO can achieve a lowering of $NO_x$ and particulate matter emission, which has a positive impact on environment. This is a suitable way of accomplishing emissions below the limits of a newly introduced ban for highly-polluting, older diesel vehicles, for example in Germany. HVO has a naturally high cetane number, which is very useful for increasing lower cetane fuels, which could also be conducive to other alternative fuels for lowering emissions, as well as having sufficient fuel properties.

The production technology of this biofuel is intergenerational in its own way, as raw materials, both food and waste, can be used for its production without significantly changing the hydrotreatment conditions. It is also beneficial for this technology that it can be operated, after minor modifications, directly in existing refineries.

**Author Contributions:** P.Z., V.H., M.K., J.T., M.O., J.M., V.H., and M.P. designed methodology; P.Z., V.H., M.K., J.T., M.O., J.M., V.H., and M.P. performed the calculation and analyzed the data; P.Z., V.H., M.K., J.T., M.O., J.M., V.H., and M.P. wrote the paper.

**Acknowledgments:** This work was supported by grants of the Grant Agency of Czech University of Life Sciences Prague IGA 2018: 31150/1312/3113—Analysis of the influence of biofuels on the operating parameters of combustion engines and IGA 2017: 31150/1312/3116—The Examination of the Influence of Blended Biofuels on Operating Parameters of CI Engines.

**Conflicts of Interest:** The authors declare no conflict of interest.

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
