# Peer review of "Hydrotreated Vegetable Oil as a Fuel from Waste Materials"

_catalysts, doi:10.3390/catal9040337_

Round 1
Reviewer 1 Report
The authors of the current manuscript describes the properties of different fuels with and without Hydrotreated Vegetable Oil admixtures and to prove the suitability of using HVO compared to FAME. The paper is interesting, but the authors should choose to revise the paper according the proposed changes, before it is finally published in the journal.
Mayor corrections:
The manuscript require major revisions. The authors should choose to revise the paper thoroughly according the proposed changes:
This paper needs some substantial work, particularly in the introduction, which is too long and poorly focused, and should be completed with more information/references about the general state of the art.
The authors must be reference some following relevant papers:
- Production from a Novel Nonedible Feedstock, Soursop (Annona muricata L.) Seed Oil. Energies 2018, 11, 2562.
- An Improvement in Biodiesel Production from Waste Cooking Oil by Applying Thought Multi- Response Surface Methodology Using Desirability Functions, Energies 2017, 10(1), 130.
- Optimizing Biodiesel Production from Waste Cooking Oil Using Genetic Algorithm-Based Support Vector Machines. Energies, 11(11), 2995.
The manuscript that the authors have presented is very poorly structured. The reorganization of the manuscript is recommend before it can be published. In this sense, the order of the sections must be:
1. Introduction,
2. Materials and method
3. Results and discussion
4. Conclusions
The Figures 2, 3, 4, 5, 6, 7, 14, 18, 19 and 20 should be clear, improve the quality of images.
The Figure 10 may include a legend explaining symbols, and vertical or horizontal tick marks.
In Figures 10 to 18, if the graphic includes a legend that explains the symbols and the vertical or horizontal marks, it is not necessary to explain it again in the text of the manuscript.
Standards and regulatory specifications used in the present study should appear in the references.
The conclusions should be amplified and improved to clarify the results obtained in this work.
Minor corrections:
Although the paper looks essentially as a valuable contribution, still some details need to be fixed, in my opinion:
The English style requires improvements. The authors must be sure that there are no grammatical errors and that the manuscript is perfectly written in English.
The authors must be sure that the bibliographical references, figures and tables are in accordance with the format of the journal.
It would be very interesting that the authors of the manuscript include more bibliographical references and, specially, references from the journal itself.
Author Response
Author's Reply to the Review Report (Reviewer 1)
This paper needs some substantial work, particularly in the introduction, which is too long and poorly focused, and should be completed with more information/references about the general state of the art.
· Introduction chorted and changed.
The authors must be referencing some following relevant papers:
- Production from a Novel Nonedible Feedstock, Soursop (Annona muricata L.) Seed Oil. Energies 2018, 11, 2562.
· Cited as number [15]
- An Improvement in Biodiesel Production from Waste Cooking Oil by Applying Thought Multi- Response Surface Methodology Using Desirability Functions, Energies 2017, 10(1), 130.
· Cited as number [9]
- Optimizing Biodiesel Production from Waste Cooking Oil Using Genetic Algorithm-Based Support Vector Machines. Energies, 11(11), 2995.
· Cited as number [10]
The manuscript that the authors have presented is very poorly structured. The reorganization of the manuscript is recommended before it can be published. In this sense, the order of the sections must be:
1. Introduction,
2. Materials and method
3. Results and discussion
4. Conclusions
· Done
The Figures 2, 3, 4, 5, 6, 7, 14, 18, 19 and 20 should be clear, improve the quality of images.
· Done or removed
The Figure 10 may include a legend explaining symbols, and vertical or horizontal tick marks.
· Done
In Figures 10 to 18, if the graphic includes a legend that explains the symbols and the vertical or horizontal marks, it is not necessary to explain it again in the text of the manuscript.
· Done
Standards and regulatory specifications used in the present study should appear in the references.
· Cited as number [32-34]
The conclusions should be amplified and improved to clarify the results obtained in this work.
· Done
Minor corrections:
Although the paper looks essentially as a valuable contribution, still some details need to be fixed, in my opinion:
The English style requires improvements. The authors must be sure that there are no grammatical errors and that the manuscript is perfectly written in English.
· Revised by English lecturer.
The authors must be sure that the bibliographical references, figures and tables are in accordance with the format of the journal.
· Revised and updated.
It would be very interesting that the authors of the manuscript include more bibliographical references and, specially, references from the journal itself.
· Added suggested articles above.

Reviewer 2 Report
I think this paper has a valuable message and should be published after suitable revision. Some og the problems are listed below:
1. I feel the introduction is very long and contains a lot of information about the hydro treating process that should either be shortened or simply referenced. II was wondering for a while if this was a review article instead of a research paper.
2. I would like to see the methods and materials section before the results so I could be sure what experiments were done in the study.
3. Could the chromatography conditions be better specified in paragraph form? Similarly, could the samples and standard test methods be better documented in the results where tose results were gives?
4. If possible could the authors also provide ASTM methods if there there is an equivalent?
There are also areas where descriptions could be clarified. Some of these however will likely be changed based on the above comments.
Author Response
1. I feel the introduction is very long and contains a lot of information about the hydro treating process that should either be shortened or simply referenced. II was wondering for a while if this was a review article instead of a research paper.
· Introduction shortened and changed.
2. I would like to see the methods and materials section before the results so I could be sure what experiments were done in the study.
· Done
3. Could the chromatography conditions be better specified in paragraph form? Similarly, could the samples and standard test methods be better documented in the results where tose results were gives?
· Added more information.
4. If possible could the authors also provide ASTM methods if there there is an equivalent?
· Added to table with standards.
There are also areas where descriptions could be clarified. Some of these however will likely be changed based on the above comments.
· Descriptions revised.

Round 2
Reviewer 1 Report
The authors of the current manuscript describes the properties of different fuels with and without Hydrotreated Vegetable Oil admixtures and to prove the suitability of using HVO compared to FAME. The paper is interesting, but the authors should choose to revise the paper according the proposed changes, before it is finally published in the journal.
Minor corrections:
Although the paper looks essentially as a valuable contribution, still some details need to be fixed, in my opinion:
The English style requires improvements. The authors must be sure that there are no grammatical errors and that the manuscript is perfectly written in English.
In line 358: The sentence “The characteristics for diesel compared to HVO were practically the same, as illustrated by Sugiyama article [17]” should be replaced by:
“The characteristics for diesel compared to HVO were practically the same, as illustrated by Sugiyama [17]”
In line 358: The sentence “According to the article [36]…” should be replaced by:
“According to Šimáček et all [36]…”
On lines 368, 371 and 378, the word "article" should be replaced with the name of the authors of the articles referred to in the references.
Author Response
Review 1
Minor corrections:
Although the paper looks essentially as a valuable contribution, still some details need to be fixed, in my opinion:
The English style requires improvements. The authors must be sure that there are no grammatical errors and that the manuscript is perfectly written in English. After the second revision, the article was finalized and reviewed by 2 chemical engineering translators.
In line 358: The sentence “The characteristics for diesel compared to HVO were practically the same, as illustrated by Sugiyama article [17]” should be replaced by: “The characteristics for diesel compared to HVO were practically the same, as illustrated by Sugiyama [17]” DONE
In line 358: The sentence “According to the article [36]…” should be replaced by: “According to Šimáček et all [36]…” DONE
On lines 368, 371 and 378, the word "article" should be replaced with the name of the authors of the articles referred to in the references. DONE
Thank you for your comments.

Reviewer 2 Report
The rvised version of the paper is significantly better and I believe it is suitable for publication.
Author Response
Review 2
After the second revision, the article was finalized and reviewed by 2 chemical engineering translators.
Thank you for your comments.
